# Identification of an allosteric binding site on the human glycine transporter, GlyT2, for bioactive lipid analgesics

Shannon N Mostyn[1], Katie A Wilson[2], Alexandra Schumann-Gillett[2], Zachary J Frangos[1], Susan Shimmon[3], Tristan Rawling[3], Renae M Ryan[1], Megan L O'Mara[2], Robert J Vandenberg[1]*

[1]School of Medical Sciences, Faculty of Medicine and Health, University of Sydney, Sydney, Australia; [2]Research School of Chemistry, College of Science, The Australian National University, Canberra, Australia; [3]School of Mathematical and Physical Sciences, University of Technology Sydney, Sydney, Australia

**Abstract** The treatment of chronic pain is poorly managed by current analgesics, and there is a need for new classes of drugs. We recently developed a series of bioactive lipids that inhibit the human glycine transporter GlyT2 (SLC6A5) and provide analgesia in animal models of pain. Here, we have used functional analysis of mutant transporters combined with molecular dynamics simulations of lipid-transporter interactions to understand how these bioactive lipids interact with GlyT2. This study identifies a novel extracellular allosteric modulator site formed by a crevice between transmembrane domains 5, 7, and 8, and extracellular loop 4 of GlyT2. Knowledge of this site could be exploited further in the development of drugs to treat pain, and to identify other allosteric modulators of the SLC6 family of transporters.
DOI: https://doi.org/10.7554/eLife.47150.001

*For correspondence:
robert.vandenberg@sydney.edu.au

Competing interests: The authors declare that no competing interests exist.

## Introduction

Inhibitory glycinergic neurotransmission plays an important role in the spinal cord dorsal horn, regulating excitatory tone in the ascending pain pathway to prevent excess nociceptive signalling (*Todd, 2010*). Concentrations of glycine within the synapse are tightly controlled by two subtypes of secondary active glycine transporters, GlyT1 (SLC6A9) and GlyT2 (SLC6A5) (*Eulenburg et al., 2005*). GlyT1 is expressed throughout the central nervous system, while the expression of GlyT2 is localised to presynaptic inhibitory neurons and allows for rapid removal of glycine from inhibitory synapses and for recycling glycine into synaptic vesicles (*Roux and Supplisson, 2000*; *Zafra et al., 1995*). The unique role of GlyT2 in pain processing has driven the development of a number of GlyT2 inhibitors for the treatment of chronic pain (*Caulfield et al., 2001*; *Takahashi et al., 2014*; *Vandenberg et al., 2014*; *Xu et al., 2005*). Inhibition of GlyT2 in this region should increase glycine concentrations within the synapse, allow prolonged activation of glycine receptors, and reduce the firing of excitatory pain-projecting neurons (*Cioffi, 2018*).

We have previously developed a new class of GlyT2 inhibitors based on the structure of the endogenous analgesic bioactive lipid, N-arachidonyl glycine (NAGly) (see *Figure 1—figure supplement 1* and *Supplementary file 2* for representative structures) (*Mostyn et al., 2017*; *Mostyn et al., 2019*). NAGly has a relatively low potency (IC$_{50}$9 µM) (*Wiles et al., 2006*), and using a medicinal chemistry approach we prepared a number of synthetic acyl amino acids that inhibit GlyT2 at concentrations in the low nanomolar range. Bioactive lipid inhibitors containing positively charged amino acid head groups are the most potent followed by aromatic, aliphatic, polar, and negatively charged amino acid head groups. One of these lipids, oleoyl D-lysine (ODLys), is potent,

metabolically stable, blood brain barrier permeable, and produces analgesia in a rat model of neuropathic pain with minimal side effects (*Mostyn et al., 2019*). In this study, we have investigated how bioactive lipids bind and inhibit GlyT2, which will provide a structural framework for further design of allosteric inhibitors of GlyT2 that could form part of long term treatment options for patients with chronic pain.

Glycine transporters are members of the neurotransmitter sodium symporter (NSS) or SLC6 family of transporters which are secondary active transporters that exploit the Na$^+$ gradient to drive transport of amino acids and amino acid derivatives across cell membranes (*Kristensen et al., 2011*). Structures of the *Drosophila* dopamine transporter (dDAT) (*Penmatsa et al., 2013*; *Penmatsa et al., 2015*; *Wang et al., 2015*), the human serotonin transporter (hSERT) (*Coleman et al., 2016*; *Coleman et al., 2019*), and the bacterial leucine transporter (LeuT) (*Krishnamurthy and Gouaux, 2012*; *Yamashita et al., 2005*), suggest a common transport mechanism for the SLC6 family. Substrate and co-transported ions enter the extracellular facing vestibule, followed by movements of extracellular loop 4 (EL4) to close the extracellular gate. Unwound regions approximately half way across the transmembrane helices TM1, TM5, TM6, and TM7 form twisting hinges to rearrange around bound substrate and allow TM1a to swing open and release substrate into the cytoplasm (*Forrest et al., 2008*; *Kazmier et al., 2014*).

Inhibitor bound structures of dDAT and hSERT are in outward-open conformations with core TM helices preventing occlusion of the binding site and provide insight into transport-unfavourable conformations. In the nortriptyline bound dDAT structure (*Penmatsa et al., 2013*), there is a 10 Å opening compared to the occluded substrate bound structure of LeuT (*Yamashita et al., 2005*), which suggests that typical inhibitors of this family bind in the central cavity to stop transport by preventing the closure of the extracellular gate. Bioactive lipid inhibitors are structurally dissimilar from typical inhibitors and the question arises as to how they inhibit GlyT2. We recently showed, using molecular dynamics (MD), that the bioactive lipids NAGly and oleoyl-L-Carnitine (OLCarn) embedded in membranes containing GlyT2 do not perturb the biophysical properties of the bilayer, or alter the structure of GlyT2, despite being present at a concentration an order of magnitude higher than the IC$_{50}$ for OLCarn inhibition of GlyT2 (*Schumann-Gillett and O'Mara, 2019*). We have also demonstrated that, while the compounds have a high apparent affinity for GlyT2, the closely related glycine transporter GlyT1 is insensitive to the acyl amino acids (*Mostyn et al., 2017*; *Mostyn et al., 2019*). This suggests that the compounds do not cause a general disruption of the membrane, but rather their inhibitory effects are mediated by binding to a specific site on GlyT2.

The mechanism of inhibition has been investigated for NAGly and oleoyl-L-lysine (OLLys) and in both cases the lipids are not competitive (*Mostyn et al., 2019*; *Wiles et al., 2006*), suggesting the compounds bind to a separate site to that of the substrate glycine, however this site remains elusive. We have demonstrated that chimeric GlyT2 transporters containing EL4 from GlyT1 are insensitive to inhibition by NAGly and OLCarn (*Carland et al., 2013*; *Edington et al., 2009*). Furthermore, a single conservative point mutation in EL4 of GlyT2, I545L, results in transporters with reduced sensitivity to lipid inhibitors. Bioactive lipids may therefore inhibit GlyT2 by binding at a site that influences the substantial conformational changes of EL4 required for transport.

In this study, we used a mutagenesis approach in combination with ligand docking and MD simulations of a GlyT2 homology model to understand how acyl amino acid inhibitors interact with GlyT2. Our results resolve differences in structure activity for inhibitors with varying amino acid head groups, and we show that bioactive lipids bind to a novel extracellular allosteric site on the transporter.

## Results

### Screening GlyT2 point mutations

The binding site for the allosteric serotonin reuptake inhibitor, (*S*)-citalopram, lies in the extracellular facing vestibule of SERT, formed by residues from TM1b, TM6a, TM10, and TM11 (*Coleman et al., 2016*). To determine if bioactive lipids inhibit GlyT2 by binding to this 'vestibule allosteric site,' corresponding residues in GlyT2 were mutated to remove potential interactions. OLCarn contains a bulky, zwitterionic head group conjugated to an oleoyl lipid tail, and was used as a screen to test mutant transporters for sensitivity to inhibition. None of the transporters containing mutations in the

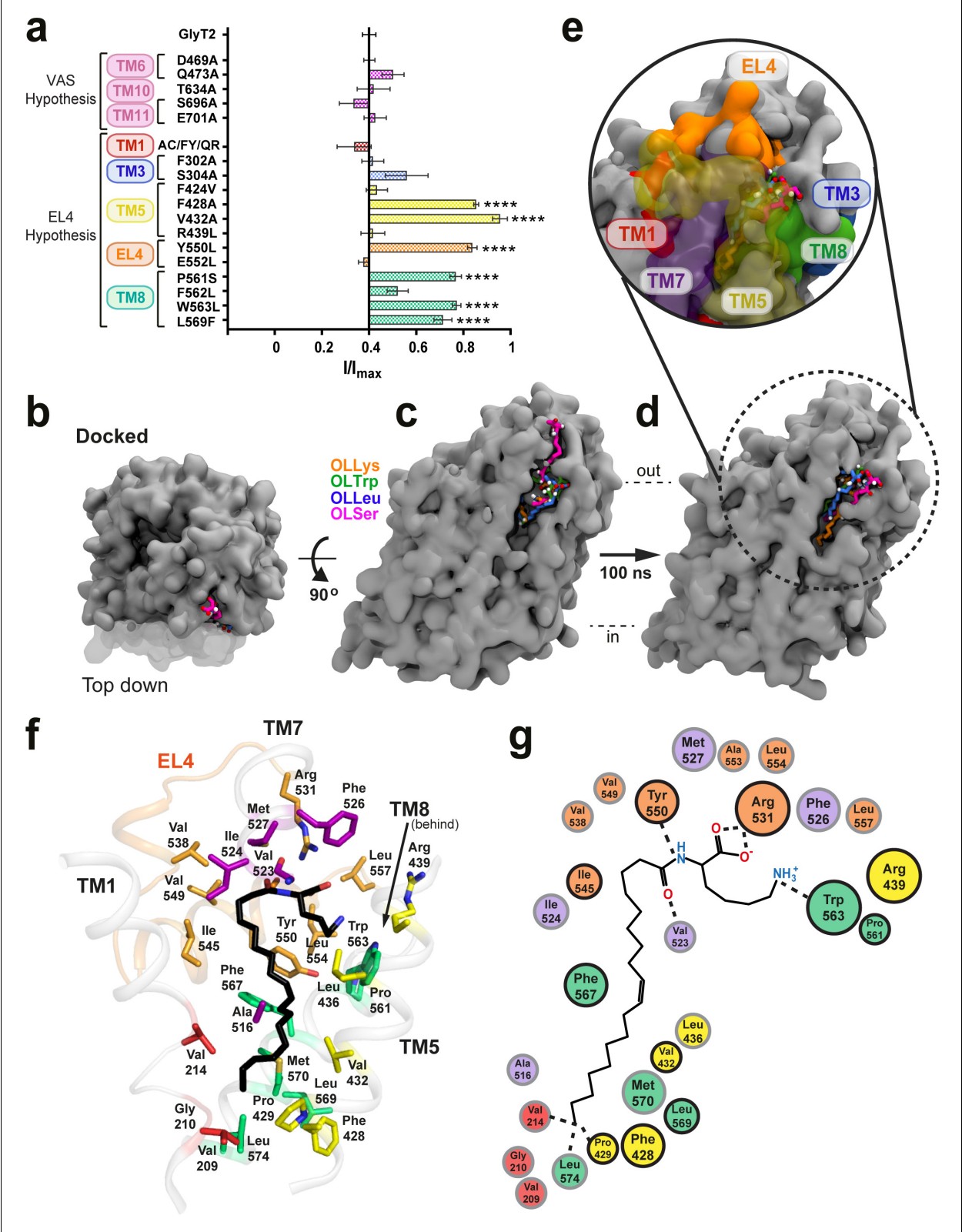

**Figure 1.** Defining the bioactive lipid binding site. (a) Inhibition of GlyT2 by 1 µM oleoyl L-carnitine. Transporters were WT or with mutations to the vestibule allosteric site (VAS) or adjacent to EL4. The reduced glycine transport currents were normalised to the current elicited by glycine alone. Data represented are means ± SEM with ****p<0.001 following a one way ANOVA test. (b–e) All lipid inhibitors docked to GlyT2 and burrowed into an area between EL4, TM1, TM5, TM7, and TM8 during 100 ns of simulation. (b) The initial docked poses, viewed top down from the extracellular side of GlyT2

*Figure 1 continued on next page*

*Figure 1 continued*

(grey surface). Sections of TM1, TM5, and TM7 are cut away in (**b–d**) to show the docking cavity (transparent surface). (**c**) The initial docked poses of OLLys (orange sticks), OLTrp (green sticks), OLLeu (blue sticks), and OLSer (magenta sticks) overlaid on GlyT2. (**d**) The conformations of the lipid inhibitors after 100 ns of unrestrained MD simulation, overlaid on GlyT2 from the OLLys simulation. (**e**) A close-up view of the inhibitors following 100 ns of simulation, with surrounding TM helices, including those cut away in panels (**b–d**), shown as a coloured surface. (**f–g**) Map of key regions in the extracellular allosteric site. (**f**) 3D arrangement of residues within 4 Å of OLLys (black sticks) following 100 ns of simulation. Residues have side chains shown as sticks, with side chains coloured TM1 (red), TM5 (yellow), TM7 (purple), EL4 (orange), TM8 (green). V523 is a backbone interaction. F428, R439, and L569 are >4 Å but shown for reference. (**g**) 2D representation, with residues studied via mutagenesis outlined in black.

DOI: https://doi.org/10.7554/eLife.47150.002

The following figure supplements are available for figure 1:

**Figure supplement 1.** Protonation states of OLLys (orange), OLTrp (green) OLLeu (blue), and OLSer (magenta) used in the docking and molecular dynamics simulations.

DOI: https://doi.org/10.7554/eLife.47150.003

**Figure supplement 2.** Location of mutated residues on GlyT2.

DOI: https://doi.org/10.7554/eLife.47150.004

**Figure supplement 3.** Sequence alignment of LeuT, hSERT, dDAT, hDAT, GlyT2, and GlyT1.

DOI: https://doi.org/10.7554/eLife.47150.005

**Figure supplement 4.** The search space for the docking to the GlyT2 homology model (grey ribbons) was defined as a rectangular box (green/red/blue shaded walls) near mutated residues on EL4, TM5, TM7 and TM8.

DOI: https://doi.org/10.7554/eLife.47150.006

**Figure supplement 5.** Resulting poses from docking to the GlyT2 homology model (grey ribbons), with EL4 highlighted in orange.

DOI: https://doi.org/10.7554/eLife.47150.007

**Figure supplement 6.** After 100 ns of simulation, GlyT2 retained a similar overall conformation when the lipid inhibitors were present (GlyT2 with OLLys - blue) or absent (GlyT2 WT - wheat), as shown.

DOI: https://doi.org/10.7554/eLife.47150.008

**Figure supplement 7.** The root-mean-square deviation (RMSD) of GlyT2 with no lipid inhibitor bound or when the lipid inhibitor is bound with tail inserted into extracellular pocket and directed towards TM5.

DOI: https://doi.org/10.7554/eLife.47150.009

**Figure supplement 8.** The root-mean-square deviation (RMSD) of GlyT2 when the lipid inhibitor is bound with the head inserted into extracellular pocket.

DOI: https://doi.org/10.7554/eLife.47150.010

**Figure supplement 9.** The root-mean-square deviation (RMSD) of GlyT2 when the lipid inhibitor is bound with tail inserted into extracellular pocket and directed towards EL4.

DOI: https://doi.org/10.7554/eLife.47150.011

**Figure supplement 10.** The root-mean-square fluctuation (RMSF) of GlyT2 diverged in specific, local regions of GlyT2 when no lipid inhibitor is bound.

DOI: https://doi.org/10.7554/eLife.47150.012

vestibule allosteric site display any change in inhibition compared to WT (*Figure 1a*). To investigate alternate sites on GlyT2 we used the observation that EL4 of GlyT2 has been shown to influence sensitivity to OLCarn and NAGly (*Carland et al., 2013*; *Edington et al., 2009*). A selection of mutations were made to residues in close proximity to EL4 that met one or a number of other criteria: residues should be accessible in the outward-facing conformation; residues which are not conserved between GlyT1 and GlyT2, and could account for differential selectivity of the inhibitors; aromatic residues, that could explain the structure activity from *Mostyn et al. (2019)*, where the most potent acyl amino acids contained positively charged or aromatic head groups; and residues that are in regions of GlyT2 that have important conformational roles in the transport cycle (residues shown in *Figure 1—figure supplements 2* and *3*). Mutations to GlyT2 residues were made either to resemble GlyT1, or to remove potential interactions with bioactive lipids but not disrupt the overall transport activity, often using substitutions present in the bacterial homologue, LeuT.

For the 'EL4 adjacent' mutations, none of the transporters containing mutations in TM1 or TM3 displayed any change in OLCarn sensitivity compared to WT. Inhibition of F424V(TM5), R439L(TM5), E552A(EL4), and F562L(TM8) transporters were also comparable to WT. Conversely, mutations to a cluster of residues on the extracellular halves of TM5 and TM8, and the neighbouring EL4b produced transporters that were less sensitive to inhibition by OLCarn, with inhibition only reaching 14.9–29.0% for F428A(TM5), V432A(TM5), Y550L(EL4), P561S(TM8), W563L(TM8), and L569F(TM8) mutants (*Figure 1a*).

For each mutant transporter with reduced sensitivity at the screening dose, concentration response curves for select acyl amino acid inhibitors were measured, with $IC_{50}$ and maximal inhibition values presented in *Supplementary file 2*. The $EC_{50}$ values of glycine for these mutant transporters are not significantly different to WT suggesting their mechanism of transport is not impaired (*Supplementary file 3*). Mutation of P429(TM5) to alanine, and F567(TM8) to alanine, valine, or leucine generated transporters that did not produce glycine dependent transport currents and were unable to be examined.

## Computational analysis of the proposed GlyT2 binding site

To further characterise the molecular basis of bioactive lipid-protein interactions, representative acyl amino acids with varying head groups were docked into our published GlyT2 homology model (*Carland et al., 2018*; *Subramanian et al., 2016*) generated from the nortriptyline bound dDAT structure (*Penmatsa et al., 2013*). The location for docking was selected based on the mutagenesis results. Specifically, OLLys, oleoyl L-Leucine (OLLeu), oleoyl L-Tryptophan (OLTrp), and oleoyl L-Serine (OLSer) (*Figure 1b–c*) were each docked into an area that encompassed the extracellular and upperleaflet embedded regions of TM5, TM8, and EL4 of GlyT2 (*Figure 1—figure supplement 4*), as this region includes the cluster of differentially sensitive residues identified using mutagenesis, that is F428 (TM5), V432 (TM5), I545 (EL4), Y550 (EL4), P561 (TM8),W563 (TM8), and L569 (TM8). The stabilities of the binding locations were assessed using unrestrained MD simulations (performed in triplicate) of the inhibitor/protein complexes, embedded in a bilayer containing POPC and 20 mol % cholesterol. As the head of the inhibitor docked in a similar location for each class of inhibitor, the positions of the inhibitors can be classified into three general poses based on the orientation of the tail: 1) head inserted into extracellular pocket, with the tail exposed to water; 2) tail inserted into extracellular pocket and the double-bond in close proximity to I545 of EL4; and 3) tail inserted into extracellular pocket, with the double-bond in close proximity to TM5 and the protein-lipid interface (*Figure 1—figure supplement 5*). Regardless of the docked position, the backbone RMSD of GlyT2 remained below 3.8 Å, indicating that GlyT2 did not display any large-scale conformational changes when OLLys, OLTrp, OLLeu, or OLSer are bound, compared to when each inhibitor was absent (see *Figure 1—figure supplements 6–10*).

Furthermore, regardless of the initial position of the inhibitor, the initial docked interaction is not maintained. In all simulations of OLLys, OLTrp, OLLeu, or OLSer which were initiated from docked poses, the inhibitor moved out of the pocket, or reoriented so that the lipid tail was directed towards TM5. Overall, MD simulations of the docked poses with the lipid inhibitor oriented headfirst in the extracellular pocket, with the tail exposed to water, or with the tail orientated so the double bond was in close proximity to I545 of EL4, indicates that neither orientation reflects stable inhibitor binding.

When the inhibitors docked with the tail directed towards TM5, the lipid inhibitors move from their initial docked pose and burrow into a novel extracellular allosteric site, with their tail wedged in a hydrophobic cavity between TM5, TM7, and TM8 (*Figure 1d–e*) and remained in that conformation for the remainder of the simulation. This was observed in all replicate simulations of OLLeu, OLLys and OLTrp, and in 2 of the three replica simulations of OLSer. It should be noted that the initial docked position of OLSer in the extracellular pocket is notably shallower than the other lipid inhibitors (*Figure 1c*). In each case when the lipid inhibitor binds in the extracellular allosteric site, the amino acid head groups of the bioactive lipids remain close to the bilayer/water interface and interact with the extracellular edges of TM5, TM7, TM8, and EL4. As the lipid inhibitors preferentially bind in this pocket, further discussion herein will focus on these simulations.

Analysis of the MD simulations shows that the binding of each lipid inhibitor to GlyT2 is mediated by several key amino acids (see *Supplementary file 4*). The lipids adopt a kinked structure with the head group interacting with a number of aromatic amino acids, as well as the side chain of R531 and backbone of V523, while the acyl tail is stabilised by aliphatic residues lining the TM5/7/8 pocket (*Figure 1f–g*).

## I545 on EL4 facilitates binding into the extracellular allosteric site

Access of lipids to the TM5/7/8 cavity is influenced by I545 in EL4 where I545 appears to sterically restrict the volume of the acyl chain binding pocket for OLLys, OLTrp, and OLSer. In all MD

simulations, the side chain of I545 remains pointed inward (e.g. *Figure 2a–b*) to interact with the acyl chain of each acyl amino acid. In this way, I545 appears to stabilise the acyl amino acid in the hydrophobic TM5/7/8 cavity, preventing the lipid from inserting into the centre of GlyT2. The beta-branched structure of I545 also facilitates the tail insertion into the TM5/7/8 cavity, with substantial curvature of the acyl chain occurring proximal to I545 (*Figures 1f* and *2b*). For OLSer, the initial

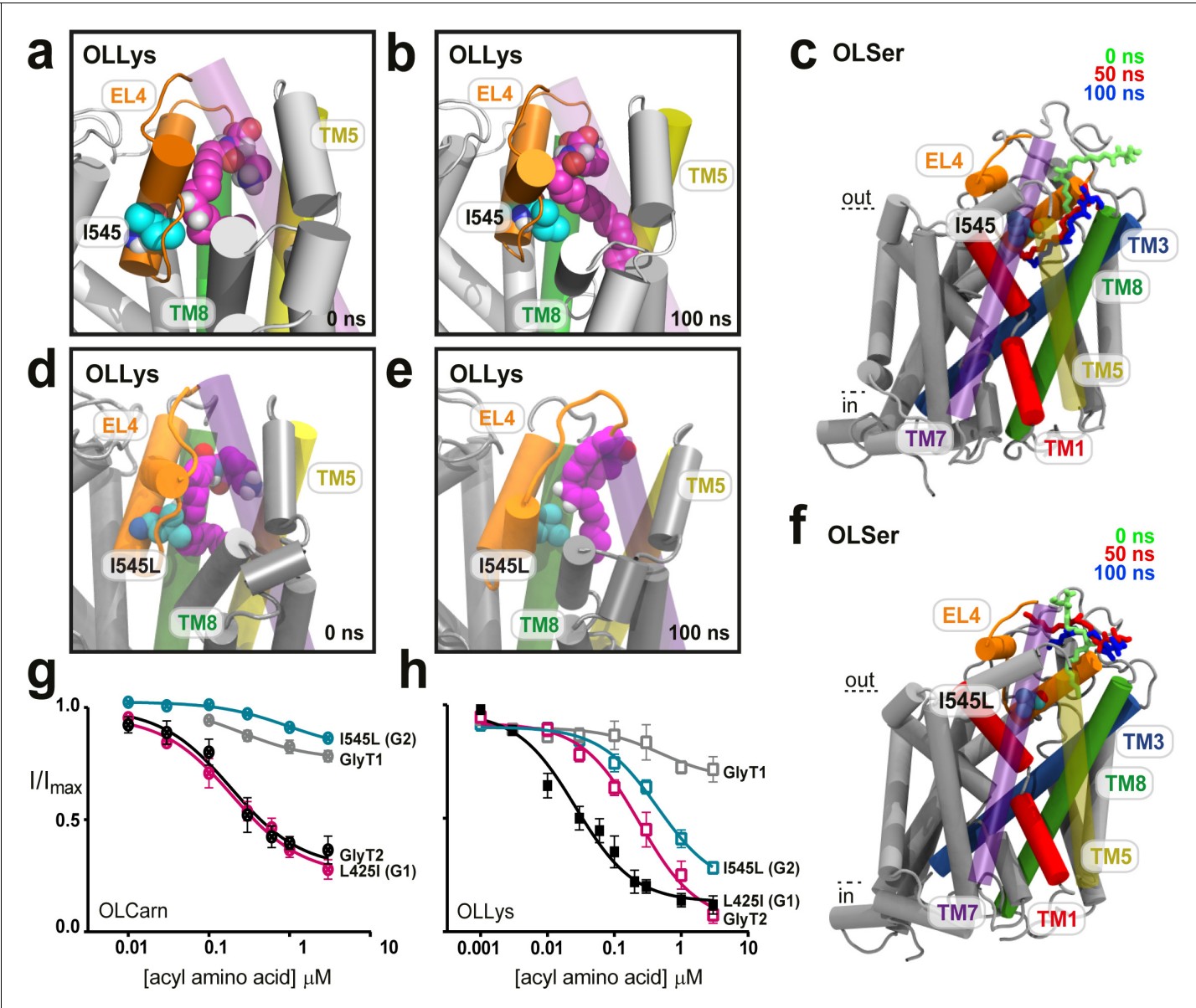

**Figure 2.** I545 facilitates lipid inhibitor binding to the extracellular allosteric site. (a) Docked position of OLLys (magenta spheres) where the acyl tail is folded at the double bond, and neighbours EL4. (b). Throughout the simulation I545 (cyan spheres) faces towards the binding cavity and interacts with the acyl tail, where there is a substantial kink adjacent to the head group. (c) Overlay showing snapshots of OLSer as it moves from the docked position (0 ns, green sticks) and into the cavity between TM5/7/8 of WT GlyT2 (50 ns, red sticks; 100 ns, blue sticks). I545 is shown as cyan spheres. TM5 and TM7 are transparent to show the cavity. (d-f) The I545L mutation (cyan spheres) sterically blocks deep insertion into the cavity. (d) Docked position of OLLys (magenta spheres) on the I545L mutant transporter and (e) following 100 ns of simulation – OLLys maintains head group interactions but the acyl tail adopts a hairpin structure to bind at a shallower cavity. (f) Overlay of OLSer as it leaves the docked position and cannot insert into the I545L mutant GlyT2, but instead idles above in the extracellular compartment. Coloured as in (c). (g-h) Acyl amino acids inhibit glycine transport currents of WT and mutant GlyT1 and GlyT2 transporters. Glycine transport currents were measured in the presence of lipids to generate concentration inhibition curves for (e). OLCarn (⊗) and (f) oleoyl L-Lys (□). GlyT2 WT is shown in black, GlyT1 WT is in grey, I545L (G2) is in teal, and L425I (G1) in pink.
DOI: https://doi.org/10.7554/eLife.47150.013

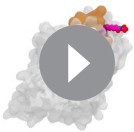

**Video 1.** OLSer binding to WT GlyT2.
DOI: https://doi.org/10.7554/eLife.47150.014

docked pose of the lipid is located at the opening to the extracellular allosteric site. OLSer burrows deeper into the binding site throughout the 100 ns simulation, facilitated by I545, which suggests insertion is an important step mediated by interactions with the acyl tail (*Figure 2c*, *Video 1*).

I545 is critical for inhibition by NAGly and OLCarn, with a conservative leucine mutation abolishing lipid inhibition (*Carland et al., 2013*; *Edington et al., 2009*). If this residue is essential to facilitate acyl amino acid burrowing into an inhibitory conformation, then its presence should be required for all bioactive lipids. To confirm this, a selection of synthetic acyl amino acids were applied to oocytes expressing the I545L mutant (*Figure 2g–h*, *Supplementary file 1*). A diverse range of bioactive lipid inhibitors (OLVal, OLAsp, or OLTrp) have reduced apparent affinities and maximal level of inhibition. For OLLys the potency is ~17 fold lower, yet 80.7% inhibition could still be attained, which suggests an inhibitory conformation can still be achieved with a leucine in this position, but the binding interaction is reduced.

GlyT2 lipid inhibitors do not inhibit GlyT1 even though the majority of residues in the extracellular allosteric site are common between GlyT1 and GlyT2, suggesting subtle substitutions (such as I545 for leucine) may impart the differential selectivity (*Figure 1—figure supplement 3*). To further investigate the role of I545, OLLys, OLLeu, OLTrp and OLSer were docked to the I545L GlyT2 mutant and simulated using the same protocol as for WT GlyT2. While the lipids initially docked in similar poses to that seen for WT GlyT2, they adopt different positions upon MD simulation. In particular, the initial docked pose of OLSer positions the lipid at the opening to the extracellular allosteric site, however steric hindrance from I545L blocks OLSer from burrowing deeper into the binding site in 2 of the three replica simulations for the I545L mutation, and instead, OLSer moves away from the extracellular allosteric site (*Figure 2f*, *Video 2*). In the single simulation where OLSer did not dissociate from the allosteric pocket of the I545L mutant, the acyl tail only partially entered the hydrophobic TM5/7/8 cavity, while the head group did not interact with the surrounding amino acids, but was partially inserted into the surrounding membrane. Conversely, OLLys is able to bind in the extracellular allosteric site of the I545L mutant GlyT2, and maintains head group interactions with residues in EL4 and TM8 (*Figure 2d–e*). However, the presence of the I545L mutation prevents full insertion of the OLLys tail into the lipid binding pocket, which instead adopts a hairpin conformation (compare *Figure 2b–2e*). This is consistent with the ability of OLLys to retain inhibition of the I545L mutant, albeit with reduced potency, and may explain how more potent inhibition can be achieved with a deeper penetrating acyl amino acid.

To further investigate selectivity between GlyT2 and GlyT1, we created the corresponding reverse GlyT1 mutation, L425I. For NOGly or OLTrp, L425I is insensitive, similar to the WT GlyT1 response; but inhibition of GlyT1 L425I by OLCarn is comparable to WT GlyT2 (IC$_{50}$195 nM; max. inhibition 74.6%). While OLLys reaches a maximal level of inhibition similar to WT GlyT2 (95.6%), the apparent affinity is 9-*fold* lower. Though clearly important, the I545 to leucine is not the only molecular determinant for bioactive lipid binding at GlyT2.

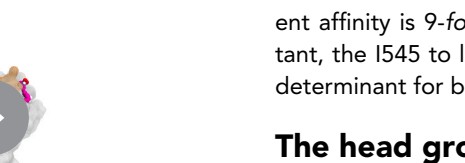

## The head group of bioactive lipid inhibitors is coordinated by aromatic residues in EL4 and TM8

Throughout the MD simulations, the amino acid head group of the bioactive lipids consistently interacts with Y550 for >94% of the total simulation time. Y550 lies approximately one turn of the

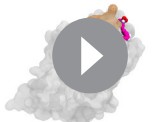

**Video 2.** OLSer binding to I545L GlyT2.
DOI: https://doi.org/10.7554/eLife.47150.015

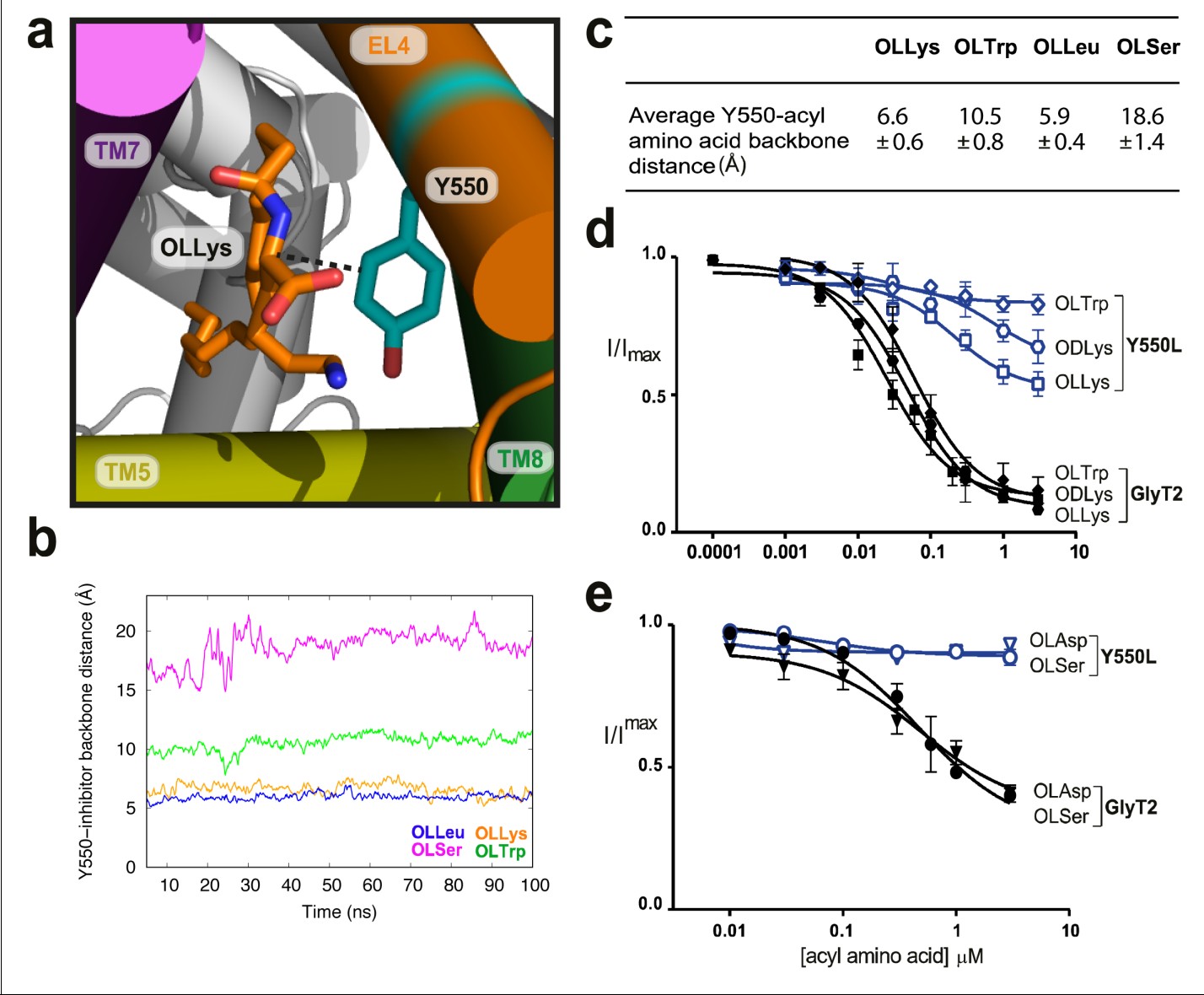

**Figure 3.** Y550 coordinates the amino acid head group. (a) Snapshot showing how the distance between C3 on the acyl-amino acids (OLLys shown here, in orange sticks) and Cα on Y550 was calculated. (b) The OLLys/OLLeu C3 (orange and blue lines, respectively) were closer to the Y550 ring than OLTrp/OLSer C3 (green and magenta lines, respectively) during the simulations, with the average distance shown in (c). (d-e) Acyl amino acid inhibition of glycine transport currents of Y550L mutant transporters. Glycine transport currents were measured in the presence of lipids to generate concentration inhibition curves for (d) OLLys (□), ODLys ( ⬡ ), OLTrp (◊), (e). OLSer (○), and OLAsp (∇). GlyT2 WT is shown in black, Y550L is in blue.
DOI: https://doi.org/10.7554/eLife.47150.016

helix up from I545 and faces away from the extracellular vestibule and into the extracellular allosteric site in the GlyT2 homology model (*Subramanian et al., 2016*). Substitution of Y550 with a leucine reduces both the apparent affinity and maximal level of inhibition for all 10 bioactive lipid inhibitors tested (*Figure 3d–e*, *Supplementary file 1*). This mutation affects acyl-amino acids with a range of different side chains suggesting an interaction is formed between the aromatic ring of Y550, and a shared moiety of the inhibitor, likely to be the amino acid backbone of the lipid head group.

The distance between the C3 of the aromatic ring of Y550 and the Cα amino acid backbone of the acyl-amino acid throughout the MD simulations is shown in *Figure 3b*. The distance ranges from 5 to 7 Å, for OLLys and OLLeu, indicating a tighter coordination for these acyl amino acids. For OLTrp the distance is ~10 Å, while Y550 does not interact with OLSer (>15 Å; *Figure 3c*).

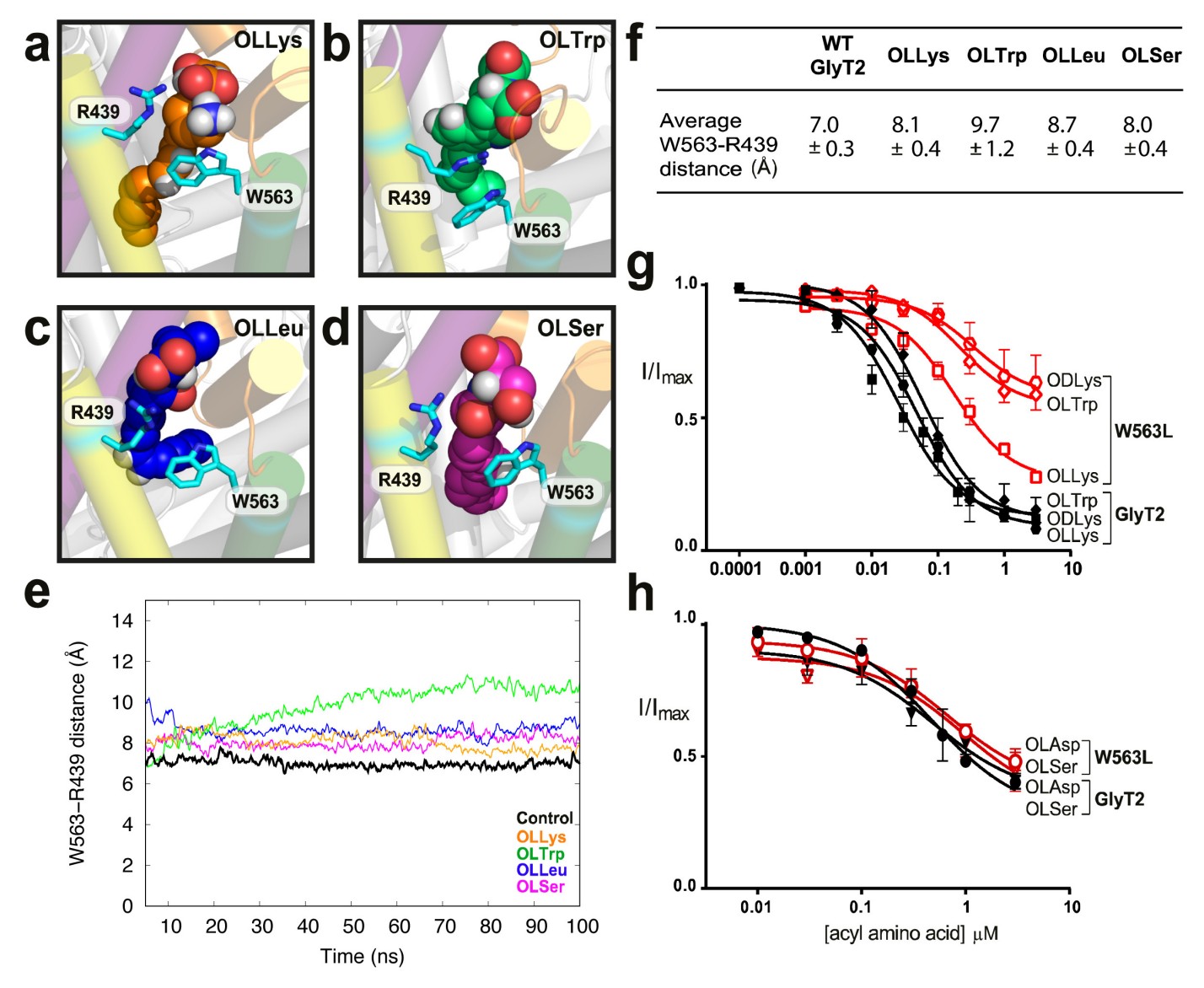

| | WT GlyT2 | OLLys | OLTrp | OLLeu | OLSer |
|---|---|---|---|---|---|
| Average W563-R439 distance (Å) | 7.0 ± 0.3 | 8.1 ± 0.4 | 9.7 ±1.2 | 8.7 ± 0.4 | 8.0 ±0.4 |

**Figure 4.** W563 and R439 act as 'gates' to stabilise the acyl amino acids in their binding cavity. (a-d) Snapshots of interactions between bioactive lipids (spheres) and W563 and R439 (cyan sticks) during the simulation. GlyT2 helices are coloured EL4 (orange), TM5 (yellow), TM7 (purple), and TM8 (green). (e) The distance between the centre of mass of W563 and R439 over 100 ns when inhibitors were bound. The control (no inhibitor) is included in black. (f) Average distances throughout the simulations. (g–h). Acyl amino acid inhibition of glycine transport currents of W563L mutant transporters. Glycine transport currents were measured in the presence of lipids to generate concentration inhibition curves for (g) OLLys (□), ODLys ( ⬡ ), OLTrp (◊), (h). OLSer (○), and OLAsp (▽). GlyT2 WT is shown in black, W563L is in red.

DOI: https://doi.org/10.7554/eLife.47150.017

As EL4 shifts into the vestibule as part of the transport cycle, it packs tightly against the core helices, including TM8, to close the extracellular gate (*Forrest et al., 2008*; *Krishnamurthy and Gouaux, 2012*). I545 and Y550 in EL4 are oriented towards the top half of TM8, near P561 and W563, and mutations of these TM8 residues differentially affect acyl amino acid inhibition. The W563L mutation has no effect on inhibition by lipids containing nucleophilic or acidic amino acid head groups, OLSer and OLAsp, while the activity of OLCarn, OLTrp, and ODLys inhibitors are all reduced compared to WT GlyT2 (*Figure 4g–h*, *Supplementary file 1*). The differential effects of the W563L mutation may be due to the contribution of π electrons for π-π and cation-π interactions with aromatic or positively charged amino acid side chains of the most potent lipid inhibitors.

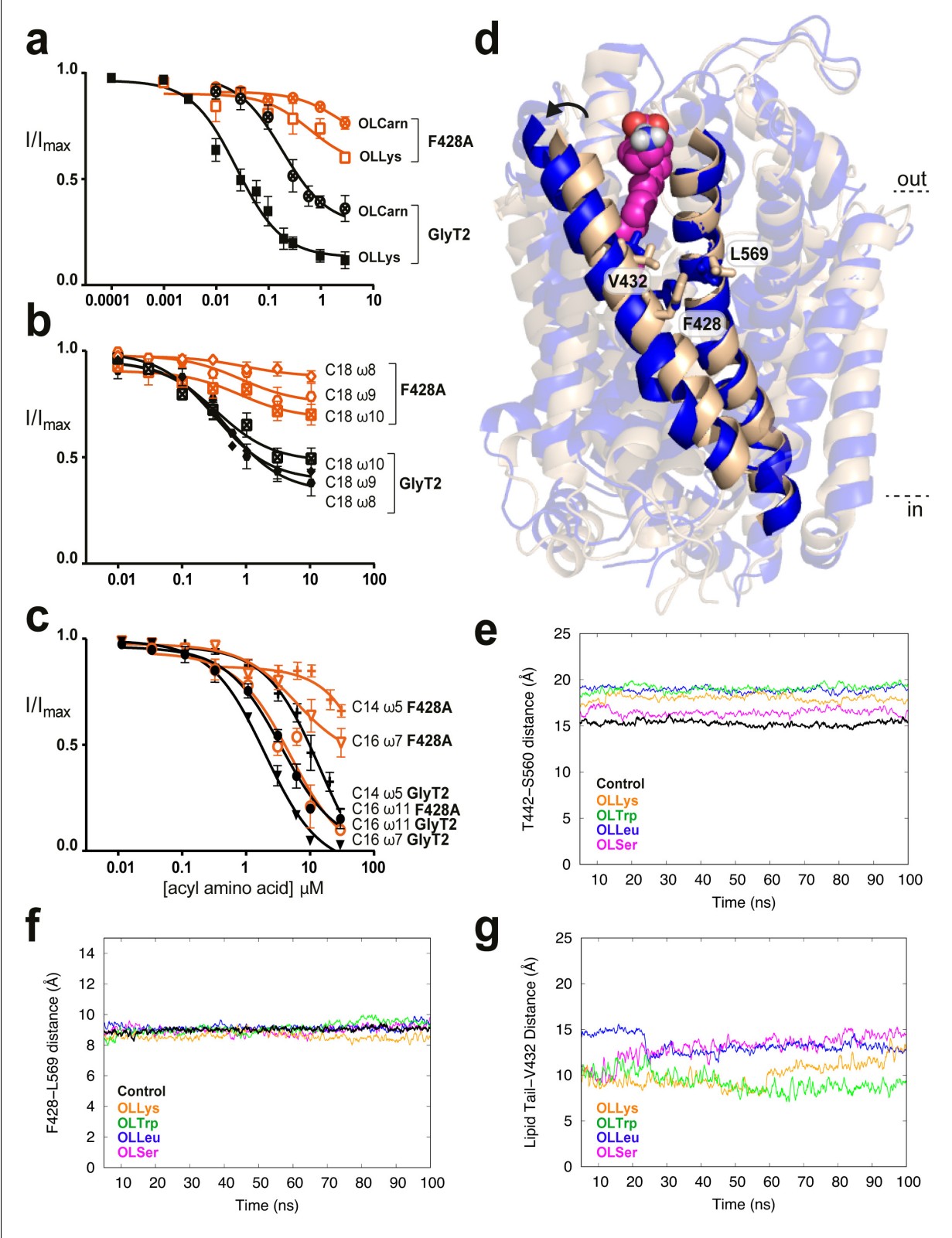

**Figure 5.** Residues in TM5 and TM8 mediate inter-helical contacts and shape the acyl tail binding cavity. (a-c) Acyl amino acids inhibit glycine transport currents of F428A mutant GlyT2 transporters. Glycine transport currents were measured in the presence of lipids to generate concentration inhibition curves for (a). OLCarn (⊗),oleoyl ʟ-Lys (□), (b) NOGly ( ⬡ ), C18 ω8 Gly (◇), C18 ω10 Gly (⊠), (c) C16 ω7 Gly (∇), C16 ω11 Gly (○) and C14 ω5 Gly (+). GlyT2 WT is shown in black and F428A is in orange. (d) Overlay of GlyT2 in the absence of inhibitor (wheat helices) with OLLys (magenta spheres)
*Figure 5 continued on next page*

*Figure 5 continued*

bound at 100 ns (blue helices). TM8 regions were aligned to show the relative movement of TM5. (**e**) Distances between the extracellular edges of TM5 (T442) and TM8 (S560) calculated from their Cα. (**f**) Distances between the middle of TM5 (F428) and TM8 (L569) calculated from their Cα. (**g**) Distances between the last carbon of the lipid inhibitor tail and the bottom of the extracellular allosteric pocket (Cα of V432).

DOI: https://doi.org/10.7554/eLife.47150.018

Molecular dynamics simulations show that W563 is particularly important for stabilising head group interactions in the binding site. In control simulations without the acyl amino acid inhibitors, the side chain of W563 associates with the R439 side chain through a cation-π interaction, forming a physical barrier, or 'gate', which limits the volume and accessibility of the cavity between TM5 and TM8. When lipid inhibitors are present, the distance between W563 and R439 remains between $7.0 \pm 0.3$ Å and $9.7 \pm 1.2$ Å (*Figure 4f*), however the precise nature of the interaction between W563 and R439 is dependent upon the amino acid head group of the bioactive lipid. OLLys binding changes the interaction between TM5 and TM8, where the indole group of W563 instead prefers to interact with the OLLys side chain for 71% of the total simulation time (*Figure 4a*). For OLSer, the polar hydroxymethyl group in the side chain also inserts between the R439 and W563 side chains as shown in *Figure 4d* and interacts with these residues for >50% of the total simulation time. The hydrophobic OLLeu side chain is within 4 Å of W563 and R439 for >90% of the total simulation time. However OLLeu does not directly interact with these residues and instead faces towards the centre of GlyT2 (*Figure 4c*). In the case of OLTrp however, the aromatic side chain forms a cation-π planar stacking arrangement, where the guanadinium group of R439 is sandwiched between OLTrp and the aromatic ring of W563 (*Figure 4b*). Notably, the interaction between R439 and OLTrp persists for the total simulation time. While the potency of OLTrp is greatly affected by the W563L mutation, inhibition of the R439L mutant is unchanged compared to WT (*Supplementary file 1*). This suggests that W563 is more important for governing head group interactions, or that there are compensatory interactions when the postive charge of R439 is lost.

For the P561S mutant transporter, inhibition by NOGly and OLVal is comparable to WT GlyT2, whereas lipids with larger or more sterically restricted head groups (OLCarn, OLTrp, and ODLys) have reduced apparent affinities (*Supplementary file 1*). P561 is the first residue at the top of TM8, where the helix breaks into a small unwound region before EL4 begins. Mutation of P561 may therefore alter the connection between TM8 and EL4, extending the helix and changing the shape of the gap between the two domains. This could limit the pocket size available to acyl amino acid inhibitors with larger head groups.

## Acyl tail binding cavity

F428 (TM5), V432 (TM5), and L569 (TM8) are located approximately in the middle of their respective transmembrane helices in a highly hydrophobic region of GlyT2. While V432A displays reduced sensitivity to bioactive lipid inhibitors with a range of acyl tails, both the L569F and F428A mutations show reduced inhibition by acyl amino acids containing the oleoyl (C18 ω9) tail, but have no effect on NAGly, which contains a polyunsaturated C20 arachidonyl tail (*Supplementary file 1*). This effect is most pronounced for F428A, and so this mutant was further explored using a range of glycine conjugated lipids with acyl chains that varied in their length and position of the monounsaturated double bond (*Mostyn et al., 2017*).

All C18 acyl-glycine inhibitors have reduced apparent affinities for the F428A transporter. A comparison between acyl-glycine analogues with double bonds in the ω8, 9, and 10 positions shows a striking relationship demonstrating that a double bond in ω8 position is markedly affected whilst a double bond in the ω10 position is only mildly affected by the mutation (*Figure 5b*). For C16 acyl-glycine inhibitors, the activity is also altered by the position of the double bond; the ω11 compound has activity on F428A comparable to WT GlyT2, but potency and maximal inhibition of the lipid with a double bond in the ω7 position is considerably reduced ($IC_{50}$ 5.8 μM, maximal inhibition 56.5%). The sensitivity of F428A to the C14 acyl-glycine is also reduced ($IC_{50}$ >30 μM) (*Figure 5c*). The compounds that have the most marked reduction in activity at F428A transporters all possess a double bond the approximate same distance from the head group, which would create a kink that needs to be accommodated by a particular cavity shape.

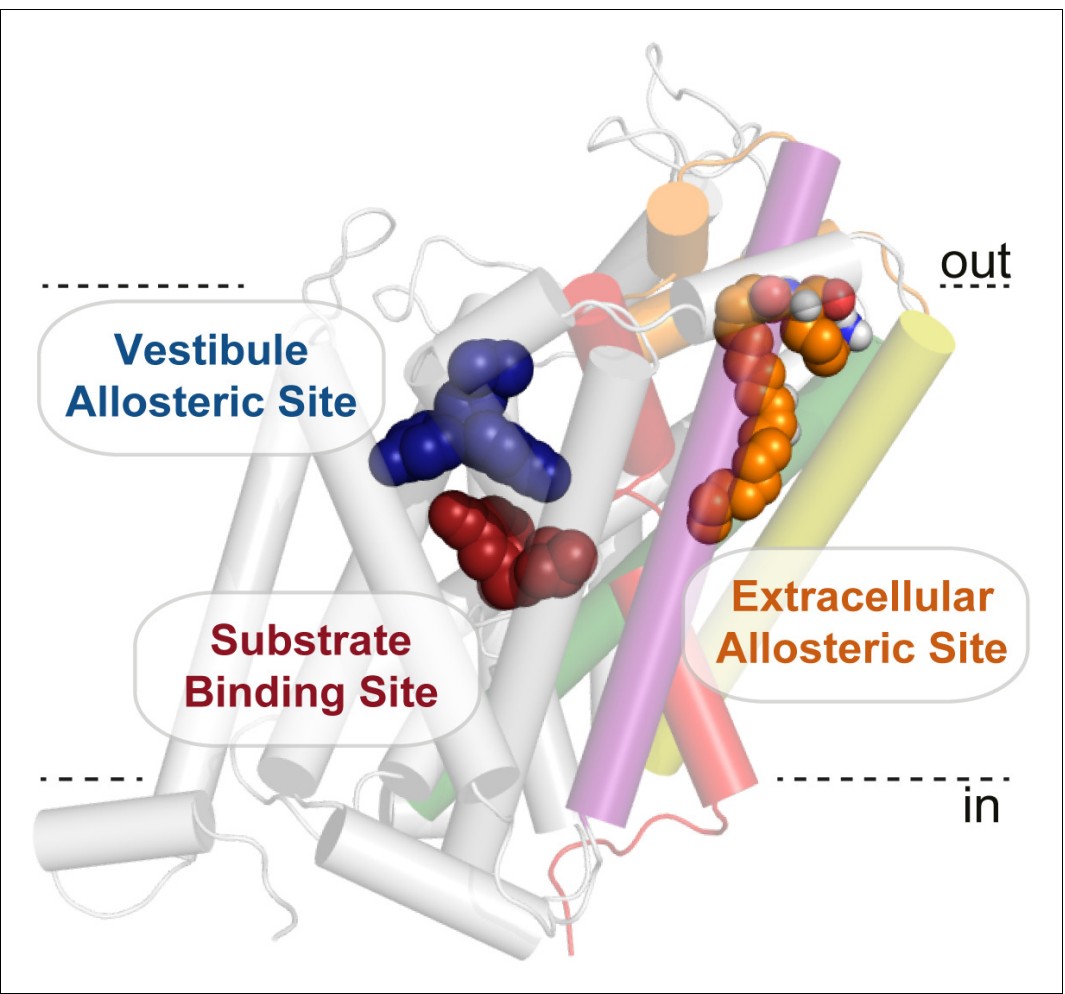

**Figure 6.** Bioactive lipid inhibitors bind to GlyT2 at an extracellular allosteric site, separate from the 'classical' central substrate and vestibule allosteric binding sites. OLLys (orange spheres) is bound to GlyT2 (100 ns). The binding location of s-citalopram at the substrate (maroon spheres) and vestibule allosteric (blue spheres) sites is superimposed from the serotonin transporter structure (PDB: 5173). GlyT2 is shown as transparent cartoon, with selected regions coloured: TM1 (red), TM5 (yellow), TM7 (purple), TM8 (green), and EL4 (orange).
DOI: https://doi.org/10.7554/eLife.47150.019

The previously mentioned MD simulations initiated from docking of four acyl-amino acid inhibitors reveal that V432 lies at the base of the lipid binding pocket facing towards the acyl tail, while F428 and L569 flank the pocket and are exposed to the centre of the hydrophobic core of the lipid bilayer. Importantly, F428 (TM5) and L569 (TM8) contact each other to mediate the TM5/TM8 helix-helix interaction. Upon insertion of the lipid inhibitors into the extracellular allosteric site, the extracellular regions of TM5 and TM8 shift apart, moving ~4 Å after 100 ns simulation (*Figure 5d–e*). F428 and L569 however, remain tightly associated to maintain contacts at the mid-point between these helices, and create a snug binding cavity for the acyl tail (*Figure 5f*). Indeed, the terminal carbon of the lipid inhibitor tail remains within 15 Å of V432 throughout the simulation (*Figure 5g*). Mutation of these inter-helical contact residues may therefore select for certain tails through changes in the volume, shape and acyl chain accessibility of the binding pocket.

## Discussion

Acyl amino acids are a new class of glycine transport inhibitors that have analgesic effects in rodent models of neuropathic and inflammatory pain with minimal overt side effects (*Mostyn et al., 2019*;

*Succar et al., 2007*; *Vuong et al., 2008*). In this study, we have explored how these compounds bind to, and inhibit, the glycine transporter, GlyT2. The acyl amino acids consist of two distinct elements: an unsaturated acyl tail and the amino acid head group. Both elements are essential for inhibition, with the length of the tail, the position of the double bond, and the chemical nature of the amino acid all influencing potency and efficacy (*Mostyn et al., 2017*; *Mostyn et al., 2019*). By studying the effects of a series of mutant transporters in TM5, TM8, and EL4, in combination with docking studies and MD, we have been able to identify structural elements in the transporter that determine head group and acyl tail specificity.

Mutations of I545 and Y550 reduce the inhibitory action of all inhibitors, suggesting that these residues interact with common features of the bioactive lipids. MD simulations suggest that I545 plays a role in steering the acyl tail into the TM5/7/8 pocket, and that a conservative mutation of this residue can cause a striking restriction of binding. Additionally, Y550 can coordinate the amino acid backbone of the bioactive lipid head group. In MD simulations, the head groups of the lipid inhibitors are stabilised by an aromatic cage formed by residues W563 (TM8), F526 (TM7), and Y550 (EL4) as well as the positively charged R439 (TM5), and R531 (EL4) which has previously been shown to be important for inhibition (*Carland et al., 2013*; *Edington et al., 2009*). Mutation of W563 has the greatest effects on inhibitors with positively charged or aromatic amino acid head groups whilst producing no effect on the efficacy of lipids containing aliphatic or uncharged polar head groups; this suggests the indole ring of W563 provides additional contacts for the side chains of the most potent inhibitors.

The stability of the extracellular allosteric site relies on inter-helical connections between EL4-TM8 and TM5-TM8. The size and configuration of the amino acid head group accommodated in this pocket could be altered with mutation to P561, an important helix breaking residue at the top of TM8. Similarly, F428 (TM5) and L569 (TM8) lie just outside the base of the cavity and form inter-helical contacts. We predict that mutation of these inter-helical contact residues may change the shape and volume of the acyl binding pocket, resulting in mutant transporters with altered head-specific and tail-specific sensitivity.

In each of the MD simulations, the bioactive lipids were docked into the region identified in the mutagenesis studies, and then allowed to find their optimal binding location following 100 ns of simulation. OLLys, OLTrp, and OLLeu were docked into the extracellular allosteric site and remained bound in the cavity for the full 100 ns of simulation. However, for OLSer, the lipid initially binds with the acyl tail partially buried within the cavity with the head group poking out to explore both the membrane and extracellular space. After 100 ns of simulation, OLSer migrates into GlyT2 (*Video 1*) to form a more stable interaction where the serine head group interacts with TM5, TM8, and EL4, as seen with other acyl amino acids. In each case, the acyl tails of bioactive lipids penetrate the transporter to forge a deeper cavity, driving apart TM5 and TM8 (*Figure 5d–e*). For the I545L transporter, bioactive lipids either did not bind in the extracellular allosteric site, or remained loosely associated in a shallow pocket with the acyl tails in a hairpin (*Figure 2e–f*, *Video 2*). Therefore, we propose that the formation of this deep binding pocket is unique to GlyT2, and may explain the selectivity of inhibition over GlyT1.

The identification of the acyl amino acid binding site by mutagenesis and MD of lipid docking raises questions about the mechanism of inhibition, and whether such a cavity exists in other closely related neurotransmitter transporters. In the transition from the outward-occluded state to the inward-open state in the bacterial homologue of the SLC6 family (LeuT), TM5 and TM7 undergo substantial conformational changes causing these helices to bend, and EL4 to pack into the central vestibule to close off the extracellular pathway (*Forrest et al., 2008*; *Krishnamurthy and Gouaux, 2012*). Furthermore, in both LeuT and the multi-hydrophobic amino acid transporter, MhsT, the intracellular half of TM5 unwinds via a conserved Gly-X$_9$-Pro motif during the transition to the inward open state to allow release of Na$^+$ ions and substrate (*Malinauskaite et al., 2014*; *Zeppelin et al., 2018*; *Stolzenberg et al., 2017*). Binding of a bioactive lipid to the extracellular allosteric binding site formed by TM5, TM7, TM8, and EL4 may therefore restrict the movements of EL4 and/or the unwinding of TM5 to inhibit the transport mechanism. Perturbation of this region via lipid binding may alter the dynamics of GlyT2, and could be a potential mechanism of inhibition that would slow, but not completely block, transport as in the case of a partial inhibitor.

Cholesterol has also been demonstrated to modulate DAT and SERT. A crystal structure of dDAT revealed a cholesterol molecule coordinated in an inner-leaflet, membrane exposed, cleft between

TM1a, TM5, and TM7 (*Penmatsa et al., 2013*). Superimposition of the dDAT structure with the inward-open structure of LeuT (*Krishnamurthy and Gouaux, 2012*) reveals a potential regulatory mechanism for cholesterol, where cholesterol may be an endogenous modulator of the dopamine transporter by inhibiting the transition to an inward-facing conformation, which has since been supported using molecular dynamics (*Zeppelin et al., 2018*). The allosteric site for cholesterol is an inner leaflet, membrane exposed site on the surface of dDAT, whereas the bioactive lipid site identified in this study is buried between helices. The lipid binding site on GlyT2 is also distinct from the central substrate binding or vestibule allosteric sites (*Figure 6*, *Figure 1—figure supplement 3*), and represents a novel extracellular allosteric site for the SLC6 family of transporters.

With the exception of I545, the majority of residues that have a significant effect on the activity of acyl amino acids are conserved among many SLC6 transporters, which suggests major features of the binding site are common. It is striking that the reverse L425I mutation in GlyT1 is sufficient to impart sensitivity of some of the inhibitors, which suggests the binding site must be mostly preformed in GlyT1, and that subtle conformational differences may accommodate different bioactive lipids. This also suggests that it may be possible to exploit these subtle differences in the structure of the extracellular allosteric site between other SLC6 members to design novel compounds that inhibit neurotransmitter transporters in a fundamentally different manner to the classical transport inhibitors such as citalopram, cocaine, or nortriptyline.

# Materials and methods

### Key resources table

| Reagent type (species) or resource | Designation | Source or reference | Identifiers | Additional information |
|---|---|---|---|---|
| Gene (human) | GlyT2a WT | UniProtKB - Q9Y345 SLC6A5 | | *Morrow et al., 1998* |
| Gene (human) | GlyT21b WT | UniProtKB - P48067 SLC6A9 | | |
| Biological sample (female *Xenopus laevis*) | Oocytes | Nasco, Wisconsin, USA | RRID:XEP_Xla100 | |
| Recombinant DNA reagent | pOTV | Krieg PA, Melton DA (1984) Functional messenger RNAs are produced by SP6 in vitro transcription of cloned cDNAs. Nucleic Acids Res 12:7057–7070 | | |
| Sequence-based reagent | Oligonucleotide primers | Sigma Aldrich (Sydney, Australia) | Primer designs were generated using https://nebasechanger.neb.com/ | |
| Commercial assay or kit | Q5 site-directed mutagenesis kit | New England Biolabs (Genesearch), Arundel, Australia | NEB.E0552S: | |
| Commercial assay or kit | mMessagemMachine T7 RNA polymerase | Ambion (Texas, USA) | AM1344 | |
| Commercial assay or kit | GeneJet Plasmid Mini Prep Kit | Thermo Fisher Scientific | K0503 | |
| Chemical compound, drug | N-arachidonyl glycine | Sapphire Biosciences | 90051 | |
| Chemical compound, drug | N-oleoyl glycine | Sapphire Biosciences | 90269 | |

*Continued on next page*

*Continued*

| Reagent type (species) or resource | Designation | Source or reference | Identifiers | Additional information |
|---|---|---|---|---|
| Chemical compounds, drugs | acyl-amino acids | *Mostyn et al., 2019* | | |
| Chemical compound, drug | Oleoyl L-carnitine | Larodan | 17–1810 | |
| Chemical compound, drug | Colleganse A | Sigma Aldrich (Sydney, Australia) | 11088793001 | |
| Chemical compound, drug | Sodium bicarbonate | Sigma Aldrich (Sydney, Australia) | S5761-500G | |
| Chemical compound, drug | Tricaine | Sigma Aldrich (Sydney, Australia) | A5040-100G | |
| Chemical compound, drug | Sodium Pyruvate | Sigma Aldrich (Sydney, Australia) | P2256-25G | |
| Chemical compound, drug | Theophylline | Sigma Aldrich (Sydney, Australia) | T1633-50G | |
| Chemical compound, drug | Ampicillin | Astral Scientific | BIOAB0028-20g | |
| Chemical compound, drug | Gentamycin | Sigma Aldrich (Sydney, Australia) | G1272-10ML | |
| Chemical compound, drug | Tetracycline hydrochloride | Sigma Aldrich (Sydney, Australia) | T7660-25G | |
| Chemical compound, drug | Glycine | Sigma | 410225–50G | |
| Software, algorithm | Labchart | ADInstruments, Sydney, Australia | | |
| Software, algorithm | Pymol | Schrodinger LLC | | |
| Software, algorithm | GraphPad Prism 7 | GraphPad Software, San Diego, CA | | |
| Software, algorithm | Gromacs 2016.1 | DOI: 10.1016/j.softx.2015.06.001 | | |
| Software, algorithm | visual molecular dynamics | DOI: 10.1016/0263-7855(96)00018-5 | | |
| Software, algorithm | Autodock vina | DOI: 10.1002/jcc.21334 | | |
| Software, algorithm | gnuplot | DOI: 10.1002/jae.885 | | |
| Other | Drummond Nanoinject | Drummond Scientific Co., Broomall, PA, USA | | |
| Other | Powerlab 2/20 chart recorder | ADInstruments, Sydney, Australia | | |
| Other | Geneclamp 500 amplifier | Axon Instruments, Foster City, CA, USA | | |

## Lipids

N-arachidonyl glycine and N-oleoyl glycine were obtained from Sapphire Biosciences (NSW, Australia); and oleoyl L-carnitine was obtained from Larodan Fine Chemicals (Malmo, Sweden). All other acyl amino acids were synthesised as previously described by Mostyn, Rawling, and colleagues (*Mostyn et al., 2017*; *Mostyn et al., 2019*).

## Creation of wild type (WT) or mutant mRNA encoding glycine transporters

Human GlyT1b or GlyT2a (herein referred to as GlyT1 and GlyT2) cDNA were sub-cloned into the plasmid oocyte transcription vector (pOTV). Site-directed mutagenesis was performed using traditional PCR techniques, and sequences confirmed by the Australian Genome Research Facility (Sydney, Australia). WT and mutant plasmid DNA were linearised with SpeI (New England Biolabs (Genesearch) Arundel, Australia) and RNA transcribed by T7 RNA polymerase using the mMessagemMachine kit (Ambion, TX, USA).

## Two electrode voltage clamp electrophysiology

Oocytes were extracted from female *Xenopus laevis* frogs and detached from follicle cell containing lobes by digestion with 2 mg/mL collagenase A (Boehringer, Mannheim, Germany). Defoliculated stage V-VI oocytes were injected with 4.6 ng of cRNA encoding WT or mutant transporter (Drummond Nanoinject, Drummond Scientific Co., Broomall, PA, USA). Surgical proceedures have been approved by the University of Sydney Animal Ethics Committee (protocol 2016/970). The oocytes were stored at 16–18°C for 2–5 days in ND96 solution (96 mM NaCl, 2 mM KCL, 1 mM $MgCl_2$, 1.8 mM $CaCl_2$, 5 mM HEPES, pH 7.55), supplemented with 2.5 mM sodium pyruvate, 0.5 mM theophylline, 50 µg/mL gentamicin and 100 µM/mL tetracycline.

2–5 days following injection, glycine transport currents were measured at −60 mV using Geneclamp 500 amplifier (Axon Instruments, Foster City, CA, USA) with a Powerlab 2/20 chart recorder (ADInstruments, Sydney, Australia) and chart software (ADInstruments). All data were subsequently analysed using GraphPad Prism 7.02 (GraphPad Software, San Diego, CA).

## Concentration responses

The function of each mutant transporter was tested by measuring glycine concentration dependent transport currents in ND96 (96 mM $Na^+$) to drive transport. $EC_{50}$ values were determined using the modified Michaelis-Menten equation:

$$I = ([GIy].I_{max})/EC_{50} + [GIy] \qquad (1)$$

where I is current (nA), [Gly] is the concentration of glycine, $I_{max}$ is the current generated by a maximal concentration of glycine (300 µM) and $EC_{50}$ is the concentration of glycine that generates a half maximal current. Values are presented as mean ± SEM obtained from $n \geq 3$ cells from at least two batches of oocytes. To determine if mutations affected glycine transport, one way ANOVA tests were employed, with a Dunnett's post-hoc test used for comparison with WT GlyT2. For L425I comparison with WT GlyT1 a two-tailed t-test was used. Statistical significance were represented as $p < 0.05$ *, $p < 0.01$ **, $p < 0.001$ *** etc. in the following figures.

The majority of N-acyl amino acids are not immediately reversible, and thus inhibitor concentration responses were performed using cumulative application. Glycine was first applied to establish maximal transport current. Glycine was then co-applied with increasing concentrations of acyl amino acid in a stepwise fashion, producing individual plateau values in response to each concentration of inhibitor.

Inhibitor concentration responses were then fit by the method of least squares using:

$$Y = Bottom + (Top - Bottom)/(1 + 10^{(x - LogIC50)}) \qquad (2)$$

where X is log[acyl amino acid] (µM), Y is current normalised to glycine in the absence of inhibitor and Top and Bottom are the maximal and minimal plateau responses respectively. This equation was constrained to have the bottom value >0, but not = 0, as to capture partial levels of inhibition, and the standard hill slope −1.0. Concentration response curves were thus able to generate $IC_{50}$ values as well as % maximum (max.) inhibition values.

$IC_{50}$ values are presented as mean and 95% confidence intervals, and % max inhibition are presented as mean ± SEM. Data are from $n \geq 3$ cells from at least two batches of oocytes. Where significant inhibition was not reached, the $IC_{50}$ value is recorded as greater than the highest concentration of acyl amino acid used. As many mutants were no longer sensitive to inhibition, significance was calculated using % maximal inhibition values for each acyl amino acid. Where inhibitors were used at

least two mutants, a one way ANOVA test with Dunnett's post-hoc tests were used for comparison with WT GlyT2. Where inhibitors were only tested on a single mutant, a two-tailed T-test was used for comparison. p values are presented as $p < 0.05$ *, $p < 0.01$ **, $p < 0.001$ *** etc. in *Supplementary file 1*.

## Molecular dynamics simulations

### Molecular coordinates and topologies

The coordinates of the experimentally validated homology model of GlyT2 (in the outward-occluded conformation) were taken from *Subramanian et al. (2016)* Substrate and bound ions were excluded to determine whether acyl-amino acid interactions were competitive or non-competitive with the glycine substrate. The Automated Topology Builder and Repository (ATB) (*Koziara et al., 2014*; *Malde et al., 2011*) was used to develop united atom coordinates and parameters for oleoyl-L-Lysine, oleoyl-L-Tryptophan, oleoyl-L-Leucine and oleoyl-L-Serine. The coordinates and topologies are available for download from the ATB (oleoyl-L-lysine MoleculeID: 252919, oleoyl-L-tryptophan MoleculeID: 252930, oleoyl-L-leucine MoleculeID: 252921, oleoyl-L-serine MoleculeID: 252932). To ensure that there was no isomerisation around the *cis* double bond, the force constant related to this dihedral angle in the acyl-amino acids was adjusted from 5.86 kJ/mol/rad$^2$ to 41.80 kJ/mol/rad$^2$. Each lipid was simulated alone in a box of water for one ns prior to docking or further simulation to ensure that this bond conformation was maintained. The parameters for POPC were those developed by *Poger and Mark (2010)*, and the cholesterol parameters were obtained from the ATB (*Canzar et al., 2013*; *Koziara et al., 2014*; *Malde et al., 2011*). The protonation state for all lipids was that in which it would most likely be found at physiological pH (pH 7): POPC and oleoyl-L-Lys were zwitterions; and oleoyl-L-Trp, oleoyl-L-Leu and oleoyl-L-Ser were deprotonated as shown in *Figure 1—figure supplement 1*.

## Acyl-amino acid docking and molecular dynamics system setup

Each acyl-amino acid was docked to our GlyT2 model (*Subramanian et al., 2016*) using Autodock vina (*Trott and Olson, 2010*). The acyl-amino acids were modelled in a united-atom configuration for consistency with the GROMOS 54a7 forcefield and subsequent MD simulation, and treated as flexible (i.e., all bonds were rotatable with the exception of the *cis* double bond and amide). The docking search space on GlyT2 was defined as a box that surrounded critically important residues in TM5, 7, 8 and EL4 such as R439 (TM5), W563 (TM8), F526 (TM7) and Y550 (EL4) (*Figure 1—figure supplement 4*, that all lie within the span of the extracellular leaflet of the membrane. The number of points that were included in the x- y- and z-directions of the docking box were: 14, 14 and 24 respectively. The N-terminus of GlyT2 was capped with a neutral acetyl group and the C-terminus was capped with a neutral amine group. Following docking, the resulting poses were categorised based on their general orientations (*Figure 1—figure supplement 5*) and one pose from each category was selected via manual inspection and simulated for each inhibitor.

The simulations were performed using GROMACS version 2016.1 with the GROMOS 54A7 force field for lipids and proteins (*Abraham et al., 2015*; *Schmid et al., 2011*). Each acyl-amino acid-bound GlyT2 was embedded in a bilayer that contained 20 mol % cholesterol and 80 mol % POPC. A control system, which lacked a bound acyl-amino acid, was also simulated. In each system, the bilayer was oriented in the x-y plane. Each bilayer system was contained in a solvated rectangular box. The simple point charge model was used to describe the water molecules and 0.15 M of Na$^+$ and Cl$^-$ ions were added. The overall charge of each system was neutral. Periodic boundary conditions were applied and each system was energy minimised by employing a steepest descent algorithm. The systems were then equilibrated by performing a series of five one ns simulations, where the backbone atoms of GlyT2 were restrained using sequentially descending force constants of 1000 kJ mol$^{-1}$ nm$^{-1}$, 500 kJ mol$^{-1}$ nm$^{-1}$, 100 kJ mol$^{-1}$ nm$^{-1}$, 50 kJ mol$^{-1}$ nm$^{-1}$ and 10 kJ mol$^{-1}$ nm$^{-1}$. The coordinates from the final frame of each 10 kJ mol$^{-1}$ nm$^{-1}$ restrained simulation were used as the starting conformation for unrestrained simulations. New velocities were assigned and unrestrained MD simulations lasting 100 ns performed in triplicate for each system.

## Simulation details

The simulation conditions were the same as used by *Schumann-Gillett and O'Mara (2019)*. Briefly, the NPT ensemble was employed and the solute (acyl-amino acid-bound GlyT2 in a bilayer) and solvent were separately coupled to an external temperature bath at 300 K. The Bussi-Donadio-Parrinello velocity rescale thermostat (*Bussi et al., 2007*) was used and the coupling constant was $\tau_T =$ 0.1 ps. The system was weakly coupled to an external pressure bath using the Berendsen thermostat, which was set to 1 bar. Semi-isotropic pressure coupling was employed, isotropic in the plane of the bilayer (x-y). The coupling constant was $\tau_T = 0.5$ ps, with an isothermal compressibility of $4.5 \times 10^{-5}$ bar. The LINCS algorithm (*Hess et al., 1997*) was used to constrain the covalent bond lengths of the solute and the SETTLE algorithm was used to constrain the geometry of the water molecules (*Miyamoto and Kollman, 1992*). The electrostatic and non-bonded interactions were updated every time step. Particle mesh Ewald summation was used to calculate the electrostatic interactions. The Lennard-Jones interactions were calculated using a 1.0 nm cut-off. Images of the docked and simulation systems were produced using the Visual Molecular Dynamics software (*Humphrey et al., 1996*), or PyMOL (The PyMOL Molecular Graphics System, Schrödinger, LLC).

## Analysis

All data were calculated using 1000 frames, corresponding to 100 ns of simulation. Averages were calculated over all frames and replicates.

### Root-mean-square fluctuation

To calculate the root-mean-square fluctuation (RMSF; i.e. standard deviation) of the backbone C$\alpha$ atoms of GlyT2, for each system the protein was fit to the first frame of the unrestrained 100 ns trajectory, and the deviation of the C$\alpha$'s was calculated using the GROMACS *gmx rmsf* tool.

### Root-mean-square deviation

To calculate the root-mean-square deviation (RMSD) of the backbone atoms of GlyT2, for each system the protein was fit to the first frame of the unrestrained 100 ns trajectory, and the deviation of the backbone was calculated using the GROMACS *gmx rms* tool.

### Pairwise distances

All pairwise distances were calculated using the GROMACS *gmx distance* tool. For the pairwise distance measurements between Y550 and the acyl-amino acids, C3 and C$\alpha$ were chosen as the respective reference groups. The distance between the centres of mass was calculated for W563 and R439. The mean distance and standard deviation for each system is given in *Figures 3c* and *4f*.

## Acknowledgements

This work was supported by the Australian National Health and Medical Research Council Project Grant APP1144429. SM was supported by an Australian Postgraduate Award, AS was supported by a Westpac Scholars Trust Future Leaders Scholarship and ZF is supported by a Research Training Program stipend. We are grateful for administrative support provided by Cheryl Handford. We are grateful for preliminary experimentation and discussions with Amelia Edington, Jane Carland, Robyn Mansfield, Lachlan Munro, Chris Sirote, Emily Crisafulli, Michael Thomas, Diba Sheipouri, and Casey Gallagher.

## Additional information

### Funding

| Funder | Grant reference number | Author |
|---|---|---|
| National Health and Medical Research Council | APP1144429 | Tristan Rawling<br>Renae M Ryan<br>Megan L O'Mara<br>Robert J Vandenberg |

| Department of Education, Employment and Workplace Relations, Australian Government | Australian Postgraduate Award | Shannon N Mostyn |
| Westpac Bicentennial Foundation | Westpac Scholars Trust Future Leaders Scholarship | Alexandra Schumann-Gillett |
| Department of Education, Employment and Workplace Relations, Australian Government | Research Training Award | Zachary J Frangos |

The funders had no role in study design, data collection and interpretation, or the decision to submit the work for publication.

### Author contributions

Shannon N Mostyn, Conceptualization; Data curation; Formal analysis; Validation; Investigation; Visualization; Methodology; Writing—original draft; Writing—review and editing; Katie A Wilson, Data curation, Formal analysis, Validation, Investigation, Visualization, Methodology, Writing—review and editing; Alexandra Schumann-Gillett, Conceptualization, Data curation, Formal analysis, Validation, Investigation, Visualization, Methodology, Writing—original draft, Writing—review and editing; Zachary J Frangos, Data curation, Formal analysis, Methodology, Writing—review and editing; Susan Shimmon, Resources, Investigation, Methodology; Tristan Rawling, Conceptualization, Resources, Supervision, Funding acquisition, Investigation, Methodology, Writing—review and editing; Renae M Ryan, Conceptualization, Funding acquisition, Investigation, Writing—review and editing; Megan L O'Mara, Conceptualization, Resources, Data curation, Software, Formal analysis, Supervision, Funding acquisition, Validation, Investigation, Visualization, Writing—review and editing; Robert J Vandenberg, Conceptualization, Resources, Data curation, Formal analysis, Supervision, Funding acquisition, Validation, Investigation, Visualization, Methodology, Writing—original draft, Project administration, Writing—review and editing

### Author ORCIDs

Robert J Vandenberg (iD) https://orcid.org/0000-0003-1523-4814

### Ethics

Animal experimentation: This study was performed in strict accordance with the recommendations in the University of Sydney Animal Ethics Committee. All of the animals were handled according to approved institutional animal care and use committee protocols 2016/970 of the University of Sydney. All surgery was performed under anesthesia, and every effort was made to minimize suffering.

### Decision letter and Author response

Decision letter https://doi.org/10.7554/eLife.47150.028
Author response https://doi.org/10.7554/eLife.47150.029

## Additional files

### Supplementary files

• Supplementary file 1. IC50 values and % max. inhibition for mutant glycine transporters.
DOI: https://doi.org/10.7554/eLife.47150.020

• Supplementary file 2. Summary of acyl amino acids, their activity on WT transporters, and their effects in MD simulations and electrophysiological recordings of mutant transporters.
DOI: https://doi.org/10.7554/eLife.47150.021

• Supplementary file 3. EC50 values for glycine transport of WT and mutant GlyT2 and GlyT1 transporters.
DOI: https://doi.org/10.7554/eLife.47150.022

• Supplementary file 4. Percentage of the total simulation time in which residues are in contact with the lipid inhibitors.
DOI: https://doi.org/10.7554/eLife.47150.023

• Transparent reporting form
DOI: https://doi.org/10.7554/eLife.47150.024

## Data availability

Simulation data (representative trajectories and starting coordinates) has been made available on Zenodo (http://doi.org/10.5281/zenodo.3355761). All other data generated or analysed during this study are included in the manuscript and supporting files.

The following dataset was generated:

| Author(s) | Year | Dataset title | Dataset URL | Database and Identifier |
|---|---|---|---|---|
| Schumann-Gillett A, Wilson KA, O'Mara ML | 2019 | Computational Data for Identification of an allosteric binding site on the Glycine Transporter, GlyT2 | http://doi.org/10.5281/zenodo.3355761 | Zenodo, 10.5281/zenodo.3355761 |

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
