## [Decision Letter]

Thank you for submitting your article "Identification of an allosteric binding site on the Glycine Transporter, GlyT2, for bioactive lipid analgesics" for consideration by *eLife*. Your article has been reviewed by three peer reviewers, one of whom is a member of our Board of Reviewing Editors, and the evaluation has been overseen by a Reviewing Editor and Richard Aldrich as the Senior Editor. The following individual involved in review of your submission has agreed to reveal their identity: Baruch Kanner (Reviewer #2).

The reviewers have discussed the reviews with one another and the Reviewing Editor has drafted this decision to help you prepare a revised submission.

Summary:

The manuscript by Mostyn et al., describes the characterization of a novel allosteric binding site in a glycine transporter, GlyT2, targeted by bioactive lipid analgesics, using biochemistry, electrophysiology, homology modeling, docking and molecular dynamics simulations. Previous studies had identified a region of the protein, extracellular loop 4, that was responsible for specificity of endogeonous lipid to GlyT2 compared to GlyT1, and had developed a class of bioactive lipid inhibitors based thereon. This study uses this information as a starting-point for docking and simulations to a homology model of GlyT2 (based on the *Drosophila* dopamine transporter, dDAT), and tests those predictions experimentally. The results provide opportunities for the synthesis of novel drugs for pain treatment. Furthermore, this work could be exploited to modulate the activity of other related neurotransmitter transporters in related allosteric sites.

Essential revisions:

1) A key conclusion of this study is that the inhibitors do not bind the known allosteric site used by citalopram. However, the authors did not perform any site-directed mutagenesis analysis on the citalopram binding site similar to that performed on the newly discovered site. Such analysis would help rule out binding to the other allosteric site.

2) An additional concern relates to the conformational changes of the ligands, which are elongated lipids with several rotatable bonds, posing a significant challenge to docking. These ligands move deeper into the pocket during molecular dynamics simulations, raising the possibility that the docking was not sufficiently robust. For example, the results might differ if a larger box were included. Therefore, important details regarding the docking analysis and careful analysis of the predicted mode of binding must be provided to support the conclusions. How were the binding site boundary identified? How were the ligands prepared for docking? How was the docking pose selected (i.e. was it the top scoring pose)? Were the top-scoring docking poses similar to each other? Using another docking method (such as a flexible docking algorithm) is an optional additional strategy that would provide alternative solutions and further support the relevance of these results.

3) The authors performed MD simulations on a single ligand-docked model. How much would the ligand insertions observed during the simulations depend on the initial conformation of the predicted complex? As described in point (2), there should have been several possible docking solutions, and each may have resulted in a different conformation after the MD simulations. Therefore, similar analysis should be carried out for more than one docking solution to see whether the final poses converge. Repeats of the MD simulations (n>=3) would also establish robustness and reproducibility. In particular, interpretations of differences between compounds should not be based on the outcome of a single trajectory (e.g. Figure 3).

4) Another concern is the potential unreliability of calculations based on a homology model in an apo conformation, i.e. without ions or a ligand in the central pocket, which is potentially unstable during MD. Completely ligand-free structures of e.g. LeuT have been shown to adopt a similar overall occluded conformation, but only after Leu99 has rotated and inserted into the central pocket. Please provide further evidence for the robustness of this choice with more detailed analysis of the stability of the system (such as RMSDs during MD) and provide some discussion of the choice. The simulation repeats mentioned in (3) should help.

5) To further establish the confidence of each predicted simulation result, please provide better quantitation of computed docking interactions, and/or visualizations that indicate the variability/spread of the results during a trajectory (e.g., using density/occupancy maps).

6) The phrase "experimentally validated", used to describe the homology model, is too vague. Some description of how the homology model was validated or why the model is accurate enough for this work is required. This should include the sequence identity of the model to the template and the corresponding expected accuracy, as well as the overall level of conservation in EL4 between the target and template.

7) The first item of the Results section (Figure 1A) shows the inhibition of glycine transport by OLCarn (the name of this compound should be given also in the Legend) in several mutants. The rationale for selecting these mutants is not clear. Several of the residues chosen are identical in GlyT2 and GlyT1, which is not sensitive to the biolipids. Moreover, it is not explained what the reason is for selecting the substitutions. For example, why is Pro-561 mutated to Ser out of 19 possible substitutions? To help clarify the choice of mutations, it might make more sense to show simulation data first and base the selection of mutations on this information. Alternatively, the authors could first analyze those residues which differ between the two GlyT's, such as the important GlyT2-I545L mutant.

8) The authors mention they identified a binding pocket unique to GlyT2 vs GlyT1, but the description is vague. A figure with explanation would help. In addition, Ile-545 of GlyT2 is a major determinant for the selectivity of the biolipids and there are steric implications when it is replaced by Leu, as in GlyT1. It would be important to put Figure 2A and B side-by side with Supplmentary Figure 8A and B and enhance visualization using, for example, stick representations.

9) The authors deduce a mechanism of OLSer binding from the docking and MD simulations (Discussion section), speculating that the "Bioactive lipids may therefore navigate the aqueous solution or interact with the cell membrane before inserting into the allosteric site, tail first". This proposed mechanism is too speculative to be included in the Discussion section, as these movements likely resulted from initial incorrect docking poses. Please revise or remove.

[Editors' note: further revisions were requested prior to acceptance, as described below.]

Thank you for resubmitting your work entitled "Identification of an allosteric binding site on the human Glycine Transporter, GlyT2, for bioactive lipid analgesics" for further consideration at *eLife*. Your revised article has been favorably evaluated by Richard Aldrich (Senior Editor) and a Reviewing Editor.

The manuscript has been improved but there are some remaining issues (or new ones that arose due to the inclusion of new data) that need to be addressed before acceptance, as outlined below:

- In subsection “Computational analysis of the proposed GlyT2 binding site”, it is stated that the initial docked position of the inhibitors is not maintained in a number of cases. Please provide evidence and quantitation for these data. Along the same lines, in subsection “Acyl-amino acid docking and molecular dynamics system setup”, it is stated that the poses were "categorized based on their general orientations and one pose was simulated". Was this by clustering or by manual inspection? Please be transparent, and quantitative where possible.

- We recommend computing the distance of the tail end to a set of points at the bottom of the cavity as a means to quantify the insertion of the lipids.

---

## [Author Response]

Essential revisions:1) A key conclusion of this study is that the inhibitors do not bind the known allosteric site used by citalopram. However, the authors did not perform any site-directed mutagenesis analysis on the citalopram binding site similar to that performed on the newly discovered site. Such analysis would help rule out binding to the other allosteric site.

Mutations were made to 6 residues in GlyT2 that correspond to 6/7 of the residues that identified in the (*s*)-citalopram allosteric binding site in the vestibule of hSERT (Coleman et al., 2016).

Mutation was not made to R216 (corresponds to R105 in SERT) which is highly conserved and forms an important structural salt bridge.

None of the mutations caused a difference in sensitivity for inhibition compared to wild type GlyT2. Data has been added to Figure 1A for comparison with mutations to the extracellular allosteric site that did have significant effects on inhibition by bioactive lipids.

Figure 1—figure supplement 2 (location of residues) and Figure 1—figure supplement 3 (sequence alignment) were also edited to include the vestibule allosteric site mutations.

2) An additional concern relates to the conformational changes of the ligands, which are elongated lipids with several rotatable bonds, posing a significant challenge to docking. These ligands move deeper into the pocket during molecular dynamics simulations, raising the possibility that the docking was not sufficiently robust. For example, the results might differ if a larger box were included. Therefore, important details regarding the docking analysis and careful analysis of the predicted mode of binding must be provided to support the conclusions. How were the binding site boundary identified? How were the ligands prepared for docking? How was the docking pose selected (i.e. was it the top scoring pose)? Were the top-scoring docking poses similar to each other? Using another docking method (such as a flexible docking algorithm) is an optional additional strategy that would provide alternative solutions and further support the relevance of these results.

Please note that, as stated in our Materials and methods section, we used Autodock vina, which is a flexible docking program that incorporates ligand flexibility during the docking process. Autodock vina implements a flexible Monte Carlo sampling optimisation with a new knowledge-based scoring function, significantly improving the quality of the prediction results compared to other flexible docking programs (Pagadala et al., 2017). While we agree with the reviewer that taking protein motion into account via a fully flexible docking algorithm would provide an alternative solution, the size of the system makes this an unrealistic choice. Instead, we specifically chose to use MD to account for the flexibility of the substrate and protein after docking. This not only allows both protein and ligand dynamics, but also recovers the local minimum energy conformation for the protein/ligand complex in an explicit solvated membrane environment, rather than simply using a scoring function.

As this was not clear in our original manuscript, we have expanded the Materials and methods section to explicitly address the reviewer’s comments by including a description of the docking region, preparation of ligands for docking and sorting of poses for subsequent MD simulation. Specifically, we state that the ligands were treated as flexible during docking and modelled with a united-atom connectivity. The box location was carefully chosen based on the mutagenesis results to position the ligand in close proximity to residues that affect the activity of GlyT2. In our revised paper we have classified our ligand binding poses into categories based on overall orientation and performed simulations on one pose from each category for each ligand, as shown in Figure 1—figure supplement 5. We believe due to the subsequent MD step, the variability within each category will be accounted for during dynamics.

3) The authors performed MD simulations on a single ligand-docked model. How much would the ligand insertions observed during the simulations depend on the initial conformation of the predicted complex? As described in point (2), there should have been several possible docking solutions, and each may have resulted in a different conformation after the MD simulations. Therefore, similar analysis should be carried out for more than one docking solution to see whether the final poses converge. Repeats of the MD simulations (n>=3) would also establish robustness and reproducibility. In particular, interpretations of differences between compounds should not be based on the outcome of a single trajectory (e.g. Figure 3).

To show the dependence of the results on the initial conformation, we have repeated all simulations so that they are performed in triplicate, after the assignment of new starting velocities to remove conformational bias.

Furthermore, we have simulated up to an additional 2 conformations from substantially different docked poses for each of the four inhibitors. All simulations presented in the paper are now performed in triplicate. Data on the persistence of contacting residues throughout the simulation time is presented as Supplementary file 4.

4) Another concern is the potential unreliability of calculations based on a homology model in an apo conformation, i.e. without ions or a ligand in the central pocket, which is potentially unstable during MD. Completely ligand-free structures of e.g. LeuT have been shown to adopt a similar overall occluded conformation, but only after Leu99 has rotated and inserted into the central pocket. Please provide further evidence for the robustness of this choice with more detailed analysis of the stability of the system (such as RMSDs during MD) and provide some discussion of the choice. The simulation repeats mentioned in (3) should help.

We have added further justification of the homology model in the results and methods sections. The stability of the protein has been clarified through the addition of RMSDs for each set of triplicate simulations for the control and for each docked pose of each inhibitor.

(Figure 1—figure supplement 7, Figure 1—figure supplement 8 and Figure 1—figure supplement 9), which complement the RMSFs (Figure 1—figure supplement 10) that were presented in the original manuscript.

5) To further establish the confidence of each predicted simulation result, please provide better quantitation of computed docking interactions, and/or visualizations that indicate the variability/spread of the results during a trajectory (e.g., using density/occupancy maps).

We have added information of the percentage time for contact residues throughout the simulation (Supplementary file 4) and added the dynamic information for the TM8-TM5 separation distance (Figure 5). We have also extended our discussion of this in the main text.

6) The phrase "experimentally validated", used to describe the homology model, is too vague. Some description of how the homology model was validated or why the model is accurate enough for this work is required. This should include the sequence identity of the model to the template and the corresponding expected accuracy, as well as the overall level of conservation in EL4 between the target and template.

As stated in response to point 4, we have added further justification of the homology model and we have added references for the experimental validation in the main text. A sequence alignment of LeuT, hSERT, dDAT, hDAT, GlyT2, and GlyT1 has been included as Figure 1—figure supplement 3.

7) The first item of the Results section (Figure 1A) shows the inhibition of glycine transport by OLCarn (the name of this compound should be given also in the Legend) in several mutants. The rationale for selecting these mutants is not clear. Several of the residues chosen are identical in GlyT2 and GlyT1, which is not sensitive to the biolipids. Moreover, it is not explained what the reason is for selecting the substitutions. For example, why is Pro-561 mutated to Ser out of 19 possible substitutions? To help clarify the choice of mutations, it might make more sense to show simulation data first and base the selection of mutations on this information. Alternatively, the authors could first analyze those residues which differ between the two GlyT's, such as the important GlyT2-I545L mutant.

The ligand docking was performed subsequent to the mutagenesis, using the location of residues which, when mutated, had a significant effect on inhibition. This way the mutagenesis data was able to define the search space for docking (Figure 1—figure supplement 4).

The rationale for mutations was rewritten to explain residues and substitutions selected.

The flow on to molecular dynamics simulations from the mutagenesis data was also rewritten to be clearer, with more emphasis placed on the I545L mutation as a starting point for mutations chosen.

8) The authors mention they identified a binding pocket unique to GlyT2 vs GlyT1, but the description is vague. A figure with explanation would help. In addition, Ile-545 of GlyT2 is a major determinant for the selectivity of the biolipids and there are steric implications when it is replaced by Leu, as in GlyT1. It would be important to put Figure 2A and B side-by side with Supplementary Figure 8A and B and enhance visualization using, for example, stick representations.

The original Supplementary Figure 8 (I545L mutation: ligand docking and molecular dynamics), was moved to Figure 2 of the main text for direct comparison with I545 wild type ligand docking and molecular dynamics.

The unique cavity has been better explained to outline the differences in the shape of the cavity and actions of the bioactive lipids when bound to either I545 wild type, and I545L mutated transporters (paragraph four of the Discussion section).

9) The authors deduce a mechanism of OLSer binding from the docking and MD simulations (Discussion section), speculating that the "Bioactive lipids may therefore navigate the aqueous solution or interact with the cell membrane before inserting into the allosteric site, tail first". This proposed mechanism is too speculative to be included in the Discussion section, as these movements likely resulted from initial incorrect docking poses. Please revise or remove.

This speculation has been removed. Further molecular dynamics studies would be needed to more accurately postulate the mechanism of binding to GlyT2.

[Editors' note: further revisions were requested prior to acceptance, as described below.]

The manuscript has been improved but there are some remaining issues (or new ones that arose due to the inclusion of new data) that need to be addressed before acceptance, as outlined below:- In subsection “Computational analysis of the proposed GlyT2 binding site”, it is stated that the initial docked position of the inhibitors is not maintained in a number of cases. Please provide evidence and quantitation for these data. Along the same lines, in subsection “Acyl-amino acid docking and molecular dynamics system setup”, it is stated that the poses were "categorized based on their general orientations and one pose was simulated". Was this by clustering or by manual inspection? Please be transparent, and quantitative where possible.

We have clarified the methods subsection “Acyl-amino acid docking and molecular dynamics system setup”.

- We recommend computing the distance of the tail end to a set of points at the bottom of the cavity as a means to quantify the insertion of the lipids.

The distances between the last carbon of the lipid inhibitor tail and the bottom of the extracellular allosteric pocket (V432) has been added as Figure 5G.